# Short-term power load forecasting based on the CEEMDAN-TCN-ESN model

**Jiacheng Huang[1], Xiaowen Zhang[1], Xuchu Jiang[1,2]\***

**1** Zhongnan University of Economics and Law, Wuhan, China, **2** State Key Laboratory of Intelligent Manufacturing Equipment and Technology, Wuhan, China

\* z0004994@zuel.edu.cn

**Data Availability Statement:** The data used in this paper are derived from the Panama national electricity load data in the Kaggle platform (https://www.kaggle.com/datasets/ernestojaguilar/shortterm-electricity-load-forecasting-panama).

## Abstract

Ensuring an adequate electric power supply while minimizing redundant generation is the main objective of power load forecasting, as this is essential for the power system to operate efficiently. Therefore, accurate power load forecasting is of great significance to save social resources and promote economic development. In the current study, a hybrid CEEMDAN-TCN-ESN forecasting model based on complete ensemble empirical mode decomposition with adaptive noise (CEEMDAN) and higher-frequency and lower-frequency component reconstruction is proposed for short-term load forecasting research. In this paper, we select the historical national electricity load data of Panama as the research subject and make hourly forecasts of its electricity load data. The results show that the RMSE and MAE predicted by the CEEMDAN-TCN-ESN model on this dataset are 15.081 and 10.944, respectively, and $R^2$ is 0.994. Compared to the second-best model (CEEMDAN-TCN), the RMSE is reduced by 9.52%, and the MAE is reduced by 17.39%. The hybrid model proposed in this paper effectively extracts the complex features of short-term power load data and successfully merges subseries according to certain similar features. It learns the complex and varying features of higher-frequency series and the obvious regularity of the lower-frequency-trend series well, which could be applicable to real-world short-term power load forecasting work.

## 1 Introduction

In recent years, the global natural environment has continued to deteriorate, energy demand has exceeded supply, and many places in the world have varying degrees of power supply shortage phenomena, but in some areas, there is a serious waste of electricity, which is a great loss to human society. Ensuring a sufficient power supply while minimizing redundant power generation is the main objective of power load forecasting; therefore, in recent years, power load forecasting has become an important direction of research in the field of natural sciences.

Power load forecasting mainly includes short-term load forecasting and medium- to long-term load forecasting. Different forecasting methods have been proposed for short-term load forecasting in recent years, which can be broadly divided into four categories according to

**Funding:** The research is supported by the Open-funding Project of State Key Laboratory of Intelligent Manufacturing Equipment and Technology (No. IMETKF2023027). The authors are grateful to other participants of the project for their cooperation.

**Competing interests:** The authors have declared that no competing interests exist.

several major areas of statistics: regression analysis methods, time series forecasting methods, machine learning methods and deep learning methods.

## 1.1 Regression analysis method

For models using the regression analysis method, the future load is forecasted based on a large amount of past electric load data, and the nonlinear relationship between the data of each period of the time series is transformed into a linear relationship by constructing a functional relationship and solving with the linear regression method.

Dudek [1] simplified the univariate short-term electric load forecasting problem by filtering out trends and seasonal variations with longer than daily variation periods, using linear regression for local modeling in the domain of the current input data, and using the stepwise regression method and the Lasso Lars method to reduce the number of explanatory variables in the regression analysis. Hirose [2] used a variable coefficient model to capture the multivariate nonlinear structure in short-term load data while using the nonnegative least squares method to estimate nonnegative regression coefficients, achieving more accurate predictions. The research method proposed by Dudek and Kei provides a basis for the daily dispatch of power grids.

Regression analysis has the advantages of high computational efficiency and strong interpretability, but due to its simple structure and heavy reliance on historical data, the robustness of its model for future prediction is poor. When unexpected events such as national holidays or severe natural disasters occur, their short-term predictions, which are based on a large amount of historical data, often deviate greatly from the actual value.

## 1.2 Time series forecasting method

Sadaei et al. [3] proposed a combination of fuzzy time series (FTS) and seasonal autoregressive integrated sliding average (SARFIMA) for forecasting seasonal time series that follow a long memory process. Mi et al. [4] proposed a method based on an improved exponential smoothing gray model for forecasting short-term power load data. Gray correlation analysis was first used to identify the main influencing factors, exponential smoothing was then applied to process the original data and build a gray forecasting model using a smoothed series in line with the exponential trend, and inverse exponential smoothing was finally used to recover the forecast values.

All the above improved time series forecasting models have achieved good forecasting results, but major limitations still exist in dealing with complex and variable nonlinear relationships among data. Generally, they can only forecast lower-frequency data and perform poorly when dealing with higher-frequency data.

## 1.3 Machine learning

Hnin and Jeenanunta [5] used support vector regression (SVR) to build a prediction model for daily electricity demand and used the particle swarm optimization algorithm (PSO) to optimize the parameters of SVR, which substantially improved the prediction accuracy of the model. Albuquerque et al. [6] used a regularized machine learning model with random forest (RF) and the Lasso Lars method, successfully extracting trends and seasonality in each time region and thus achieving more accurate forecasts in all time domains. Fan et al. [7] proposed a combined SVR-GC-RF algorithm short-term load forecasting model, using SVR to address the nonlinear relationships among data and a garbage collection algorithm (GC) to extract mutation points from long-term data and reduce randomness. Finally, the prediction performance is optimized using RF. Ribeiro [8] proposed a hybrid learning model based on a dual

decomposition approach. The locally weighted regression (STL) is used to decompose the original time series into seasonal, trend, and residual components. Then, variational mode decomposition (VMD) is adopted to decompose the STL residual into different frequencies, and seasonal naïve is used to handle the seasonal patterns. Moreover, in consideration of the non-linearities of the remaining components, extreme learning machines, ridge regression, and support vector regression models are employed to handle the STL trend and VMD components.

The accuracy of machine learning in completing prediction tasks often requires excellent feature engineering, and the setting of model parameters also has a great impact on the prediction accuracy. Machine learning models often perform poorly on chaotic systems.

## 1.4 Deep learning

In recent years, deep learning models have been increasingly adopted in the research of time series such as wind speed and temperature. Compared with traditional statistical models, deep learning models often have better performance in the prediction task of multi-feature complex time series. Lv et al. [9] proposed an effective hybrid model for wind speed forecasting tasks based on an improved hybrid time series decomposition strategy (HTD) and the novel multi-objective binary backtracking search algorithm (MOBBSA). Furthermore, the advanced Sequence-to-Sequence (Seq2Seq) predictor is used to obtain the final forecasting result. In addition, Lv et al. [10] also proposed the FWNSDEC-SSA-ConvLSTM model for enhancing the short-term wind speed forecasting performance. The filter-wrapper non-dominated sorting differential evolution algorithm incorporating K-medoid clustering (FWNSDEC) is designed to select key meteorological factors, and singular spectrum analysis (SSA) is used to decompose the meteorological factors. Then, the convolutional long short-term memory (ConvLSTM) network is adopted to obtain the final forecasting result. Similarly, deep learning is very important in the prediction task of short-term power loads.

Zhou et al. [11] proposed a novel predictive system based on a data denoising strategy, statistical predictive systems, artificial intelligence forecasting systems and a multi-objective optimization strategy. After the data denoising process, the reconstructed data are used as the input of different subsystems. Then, to obtain stable forecasting results, a multi-objective dragonfly algorithm is used to estimate the weight coefficient of the subsystems. Trierweiler Ribeiro et al. [12] used an echo state network (ESN) for short-term load forecasting and proposed a Bayesian optimization algorithm to calculate the hyperparameters of the ESN model. Guo et al. [13] used convolutional neural networks (CNNs) to cascade shallow and deep features from different scales, and the feature vectors from different scales were fused and fed into a long short-term memory network (LSTM) for future load prediction. Khwaja et al. [14] used an integrated machine learning approach based on an artificial neural network (ANN) for short-term load forecasting and proposed a technique that combines bagging and boosting to improve ANN model performance. Peng et al. [15] proposed a long short-term memory-based model for energy consumption forecasting. The empirical wavelet transform (EWT) technology was first used to decompose the energy consumption sequence into several components. Then, each component is concatenated as the input of the attention-based LSTM model to obtain the predicted value of each component. Finally, the predicted value of each component is integrated to obtain the final output.

Compared with the methods, deep learning has stronger fault tolerance and can handle complex data information. However, it is limited due to the amount of sample data needed, slow convergence, complex optimization function, and being prone to fall into local extremes.

## 1.5 Research methods proposed in this paper

This paper proposes a combined model CEEMDAN-TCN-ESN to achieve point estimate forecasting of short-term electric load. In the first stage, this paper trains and predicts the original series that have been preprocessed but not decomposed by complete ensemble empirical mode decomposition with adaptive noise (CEEMDAN) using multiple models, compares the prediction accuracy of a single model on this dataset and finds the best single prediction model. Then, the original time series of the electric load are decomposed by using the CEEMDAN decomposition technique to obtain a total of 15 subseries, including trend terms, and the same set of models is used to forecast each subseries. All 15 forecasts are added up to obtain the final forecasting results and compare the forecasting effect of a single model on the corresponding datasets before and after decomposition to reflect the enhancement effect of CEEMDAN modal decomposition on the forecasting performance. Moreover, to improve the forecasting accuracy and efficiency, this paper reconstructs all 14 subseries obtained after CEEMDAN decomposition of the original series, except the trend term, into two combined series of high frequency and low frequency.

Considering the similarity of the characteristics of lower-frequency series and trend term series, (i.e., compared to higher-frequency series, both have relatively gentle changes and more obvious regularity). Therefore, the lower-frequency series and the trend term series are combined to obtain two merged series of higher-frequency and lower-frequency trends and use the same set of models as before to model and predict the two merged series, determine the best model for each of the two series, build a combined model for prediction, and calculate the performance improvement percentage relative to the best single model when the series are not decomposed and the best single model when the series are decomposed but not reconstructed.

In this study, a special combination model is designed by combining the modal decomposition technique and the dual prediction model. Compared with the research methods without time series decomposition or using only a single prediction model, it has higher prediction efficiency and accuracy. The main contributions of this paper are as follows.

1. A special combination model is designed by combining the modal decomposition technique and the dual prediction model. Compared with the research methods without decomposition techniques or using only a single prediction model, it has higher prediction efficiency and accuracy.

2. This paper is not limited to a single traditional statistical perspective or a single machine learning perspective but considers the prediction model from two fields at the same time. The respective classical models are selected as experimental alternatives to discover the fitting effects of different classes of models on data with different characteristics. By observing their respective performances in the original series, higher-frequency, and lower-frequency-trend series to achieve a two-way comparison between horizontal and vertical.

3. In this study, four sets of experiments were conducted. The experiments include the prediction of a single model without the decomposition technique, a single prediction model with the decomposition technique but without sequence reconstruction, a single prediction model with the decomposition technique and sequence reconstruction, and a dual prediction model with the decomposition technique and sequence reconstruction.

4. A robustness analysis experiment was carried out, and the four experiments above were repeated on another dataset. The experimental results show that the proposed hybrid prediction model has a good generalization ability.

The rest of this paper is organized as follows. Complete ensemble empirical mode decomposition with adaptive noise (CEEMDAN), reconstruction, temporal convolutional network (TCN), and echo state network (ESN) are introduced in detail in Section 2. The specific learning framework of CEEMDAN-TCN-ESN will be introduced in Section 3. Section 4 contains the parameter settings of the model, the output results of the model applied to the dataset, and the performance indexes of the proposed model and the benchmark model. Section 5 applies the proposed combined model to another dataset to check the robustness of the model. Finally, the conclusions are presented in Section 6.

## 2 Methodologies

In this section, CEEMDAN, intrinsic mode function (IMF) component reconstruction, TCN, and ESN are introduced one by one, which are the basis of the combinatorial model proposed in this paper.

### 2.1 CEEMDAN

Complete ensemble empirical mode decomposition with adaptive noise (CEEMDAN) is an algorithm implemented on top of the empirical mode decomposition (EMD) algorithm.

The basic principle of the EMD algorithm is based on the local characteristic time scale of the original signal. The original signal is decomposed into characteristic mode functions to obtain a series of intrinsic mode function (IMF) components from high frequency to low frequency. The EMD algorithm has excellent properties such as adaptability and multiresolution, but the algorithm also has some obvious disadvantages, including envelope fitting bias, endpoint effect, modal blending, etc. The modal aliasing phenomenon is manifested in these two aspects: (1) signals with different feature scales appear in the same IMF component, and (2) signals with the same feature scale appear in different IMF components.

To solve the problem of modal confusion, Chang et al. and Torres et al. [16, 17] proposed two other decomposition methods, EEMD and CEEMDAN. The main improvement idea of the EEMD algorithm is that by using the zero-mean property of white noise, the white noise obeying uniform distribution is introduced several times in the process of decomposition, and the noise of the original signal itself is masked by the artificially added noise to obtain a more accurate upper and lower envelope. At the same time, the decomposition results are averaged, and the more times the averaging process is performed, the smaller the impact of noise on the decomposition.

Assuming that the original time series is $X(t)$, the decomposition steps of CEEMDAN are as follows.

(1) First, a Gaussian white noise sequence of length $t$ periods ($t$ is the length of the original time series) is added to the original sequence, and the adaptive coefficient of Gaussian white noise in each phase is set to $a_0$. Then, the first modal component $IMF_1$ is obtained by EMD decomposition of the time series after white noise has been added. Furthermore, the above procedure is repeated $p$ times to obtain $p$ $IMF_1$ and calculate their mean values to obtain $\overline{IMF_1}$. The specific process is demonstrated in Eqs (1) and (2):

$$X(t) + a_0 wn^i(\text{t}) = IMF_1^i(t) + r_1^i(t) \tag{1}$$

$$\overline{IMF_1} = \frac{1}{p} \sum_{i=1}^{p} IMF_1^i(t) \tag{2}$$

(2) Remove $\overline{IMF_1}$ from the original time series $X(t)$ and label the remaining sequence as $r_1^i(t)$. The adaptive signal $E_1(wn^i(t))$ will be calculated by EMD and added to the remaining sequence $r_1^i(t)$. Next, similar to step (1), Eqs (1) and (2) are repeated p times to obtain p $IMF_2$, and then their mean values are calculated to obtain $\overline{IMF_2}$. The specific process is shown in Eqs (3)–(5):

$$r_1(t) = X(t) - \overline{IMF_1} \tag{3}$$

$$r_1(t) + a_1 E_1(wn^i(t)) = IMF_2^i(t) + r_2^i(t) \tag{4}$$

$$\overline{IMF_2} = \frac{1}{p}\sum_{i=1}^{p} IMF_2^i(t) \tag{5}$$

(3) For the *kth* ($k = 2,3\ldots n$) modal components, similar to step (2), $\overline{IMF_k}$ is obtained. The specific process is shown in Eqs (6)–(8):

$$r_{k-1}(t) = X(t) - \overline{IMF_{k-1}} \tag{6}$$

$$r_{k-1}(t) + a_{k-1} E_{k-1}(wn^i(t)) = IMF_k^i(t) + r_k^i(t) \tag{7}$$

$$\overline{IMF_k} = \frac{1}{p}\sum_{i=1}^{p} IMF_k^i(t) \tag{8}$$

(4) The above steps will be repeated until the remaining components are no longer suitable for decomposition. Finally, all the *IMFs* that satisfy the conditions are successfully extracted, and the trend term $r_n(t)$ is obtained at the same time.

## 2.2 Reconstruction

The IMF components obtained from the CEEMDAN decomposition satisfy the local symmetry of the upper and lower envelopes with respect to the time axis. For high-frequency IMF components, the upper and lower envelopes are basically obtained by connecting numerous signal peaks, so the symmetry of the upper and lower envelopes means that the IMF component data are also basically symmetric and the mean value of the data tends to be close to zero. However, for low-frequency IMF components, due to their long signal period, the envelopes are obtained by interpolating a small number of signal peaks, resulting in a large deviation of the envelope trend from the original signal trend. Therefore, for low-frequency IMF components, the signal component data are often not symmetric when the envelope is symmetric or even far apart, and the mean value of the data generally does not tend to be close to zero [18].

Based on this, this paper first sums the IMF components obtained from CEEMDAN decomposition item by item, defines the sum of the first $i$ IMFs as $index_i$ ($i = 1,2\ldots k$), calculates the mean value of $index_1$ to $index_k$, and conducts Student's t test on whether the mean value significantly deviates from zero. The specific process is shown in Eqs (9)–(11):

$$index_i = \sum_{a=1}^{i} IMF_a \tag{9}$$

$$\overline{index}_k = \frac{1}{k}\sum_{i=1}^{k} index_i \qquad (10)$$

$$H_0 : \overline{index}_k = 0 \quad H_1 : \overline{index}_k \neq 0 \qquad (11)$$

## 2.3 TCN

The temporal convolutional network (TCN) stands for time domain short for convolutional network, which originally consisted of dilated, causal 1D convolutional layers with the same input and output lengths. The temporal convolutional network is improved based on the traditional 1D convolutional neural network while combining causal convolution, dilated convolution and residual linking. Some researchers have applied TCN model to predict short-term distance headway, ultra-short-term wind power, rockburst risk level and other things [19–21]. The convolutional structure of the TCN model is shown in Fig 1.

The receptive field of the TCN network neuron, i.e., the network memory length, is determined by the convolutional kernel size, the expansion coefficient and the number of convolutional layers, and the formula for the *t-th* operation after the expansion convolution operation is shown in Eq (12):

$$F(t) = (x*f)(t) = \sum_{i=0}^{u-1} f(i) \cdot x_{t-p\cdot i} \qquad (12)$$

In Eq (12), $x$ is the input sequence, * is the convolution operation, $u$ is the convolution kernel size, $f(i)$ is the *i-th* element of the convolution kernel, and $x_{t-p,i}$ is the element of the input sequence that corresponds to the convolution kernel.

The TCN residual module contains the underlying causal expansion convolution layer, the weight normalization layer, the activation function and the dropout layer. The structure of the N residual module is shown in Fig 2:

Weight normalization can solve the gradient explosion problem and speed up the computation effectively. The *ReLU* activation function is also used, and the dropout layer is added after

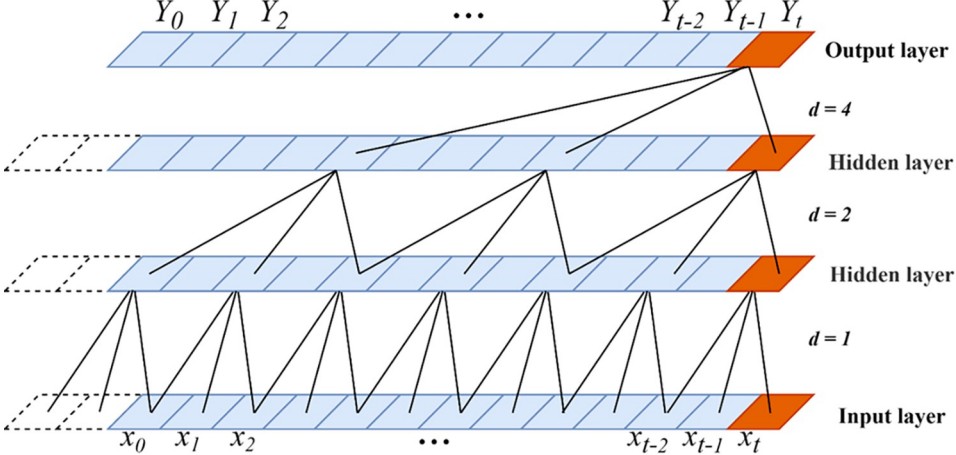

**Fig 1. The convolutional structure of the TCN model.**

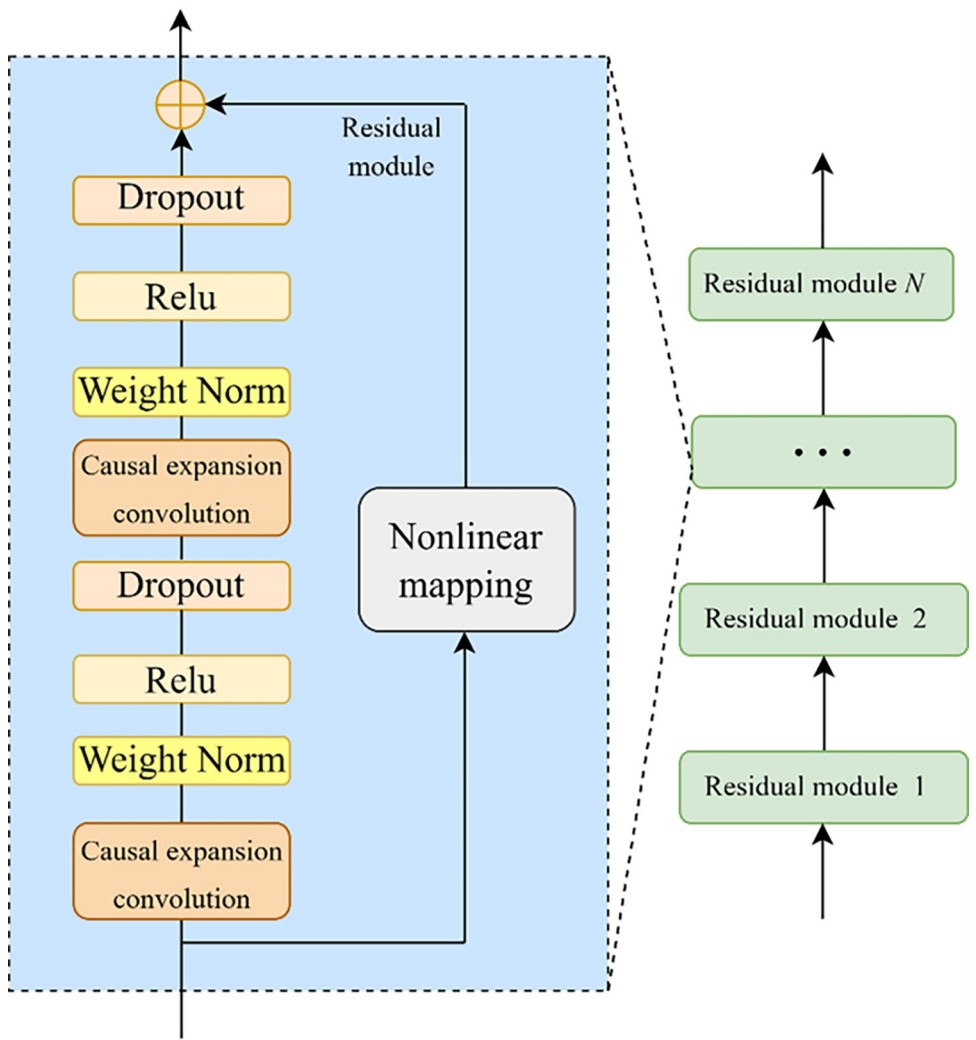

**Fig 2. The structure of the N residual module.**

the *ReLU* activation layer to prevent overfitting to achieve the regularization effect. Finally, the problem of different latitudes of the residual tensor is adjusted by nonlinear mapping (e.g., 1×1 convolution).

## 2.4 ESN

The echo state network (ESN), also known as reservoir computing, was proposed by Jaeger in 2001, and a reserve pool composed of neurons obtained by a randomly generated sparsely connected internal weight matrix is used as the hidden layer to project the input layer to a high-dimensional, nonlinear space representation. The ESN is not trained to generate the hidden layer weights of the neural network but to generate the hidden layer weights in advance, and this process is performed separately from the process of training the weights from the hidden layer to the output layer. The basic idea is based on the premise that the generated reserve pool has some good properties, i.e., it is often guaranteed that excellent performance can be obtained by training the weights from the reserve pool to the output layer using only linear methods [22, 23].

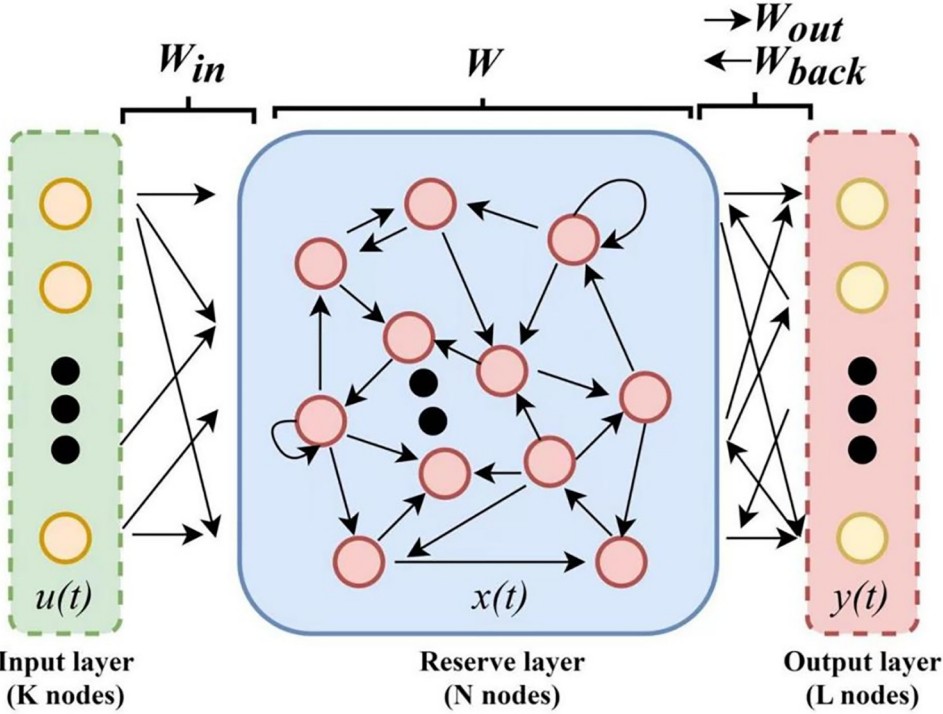

**Fig 3. The specific dissemination of the Echo State Network.**

ESN solves the problem that weights are difficult to determine in RNN by setting cyclic hidden units, fixing the input layer and hidden layer weights, and learning only the output weights, which also indirectly solves the problem of information disappearing in forward propagation and gradient explosion in back propagation in RNN.

As shown in Fig 3, the ESN consists of three basic parts, namely, the input layer, the reserve pool, and the output layer.

Input layer: Input the tensor of k×1, multiply the input by $W_{in}$ (the weight matrix with the input unit to the inside of the reserve layer) and enter the resulting product into the reserve layer.

Reserve layer (N-node network): Each node in the reserve pool corresponds to a state and is represented using $X_N$. The internal neuron connection weight matrix $W$ is also present in the reserve layer.

Output layer: Multiply the output of the reserve pool by $W_{out}$ (the weight matrix with the inside of the reserve layer to the output layer) to obtain the target value y.

The respective values of the input layer $u(t)$, reserve pool neuron hidden state $x(t)$, and output layer $y(t)$ at moment t are taken as in Eqs (13)–(15):

$$u(t) = [u_1(t), u_2(t), \dots u_K(t)]^T \tag{13}$$

$$x(t) = [x_1(t), x_2(t), \dots x_N(t)]^T \tag{14}$$

$$y(t) = [y_1(t), y_2(t), \dots y_L(t)]^T \tag{15}$$

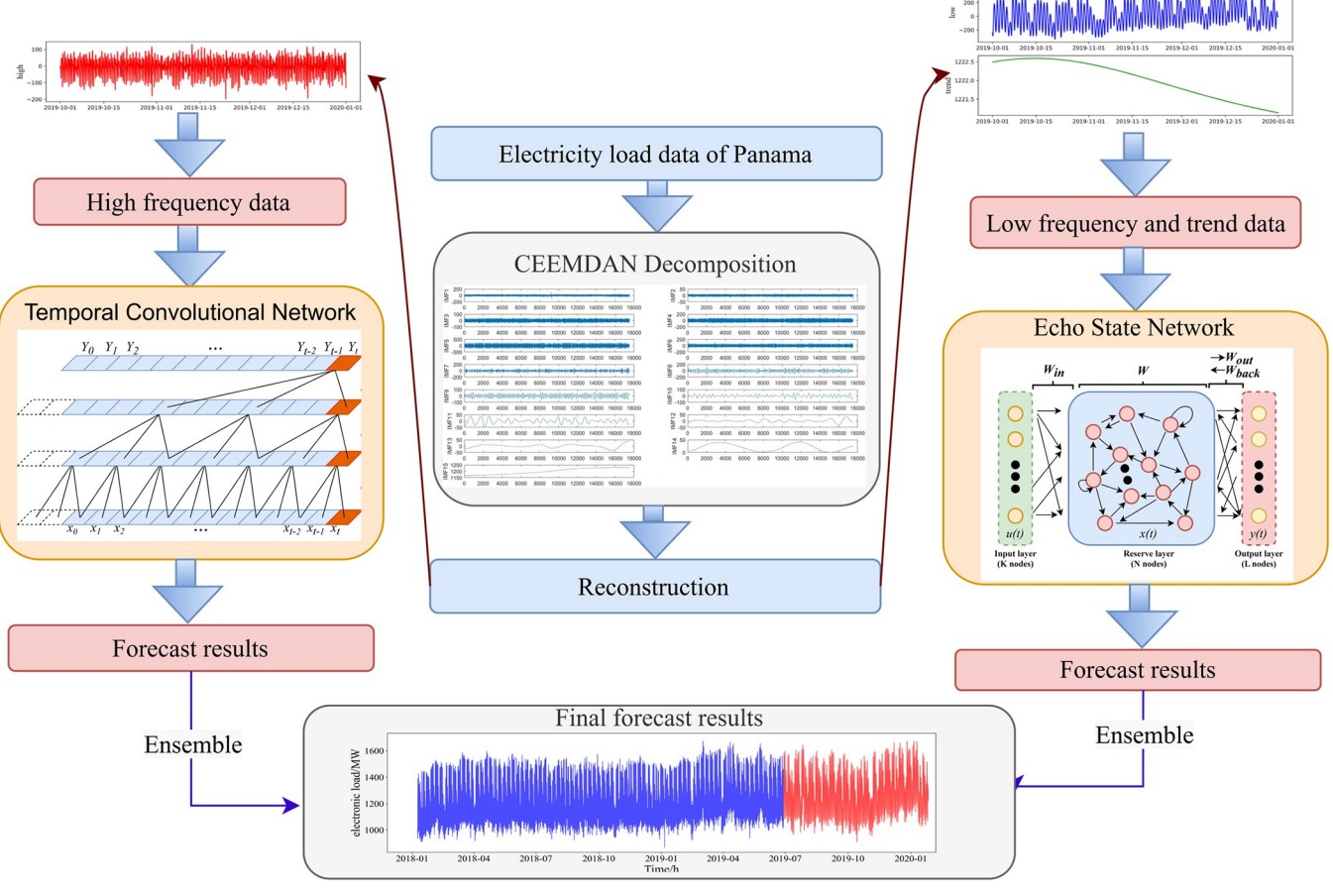

**Fig 4. Research framework.**

The specific dissemination of the Echo State Network is shown in Fig 3 and Eqs (16) and (17).

$$x(n+1) = f(W_{in}u(n+1) + Wx(n) + W_{back}y(n)) \tag{16}$$

$$y(n+1) = f^{out}(W_{out}(u(n+1), x(n+1), y(n))) \tag{17}$$

In the above equations, $f$ is the activation function of the reserve layer's cells, and $f^{out}$ is the activation function of the output layer's cells. $W_{in}$ is the connection weight matrix of the input layer to the reserve layer. $W$ is the internal connection weight matrix of the reserve layer. $W_{back}$ is the connection weight matrix of the output layer to the reserve layer. All of them are preprogrammed before the model training, and only $W_{out}$ (the connection weight matrix of the reserve layer to the output layer) is obtained through the training model.

The research framework of this paper is shown in Fig 4.

## 3 Framework of the CEEMDAN-TCN-ESN

In this section, a detailed introduction to the CEEMDAN-TCN-ESN learning framework is presented. First, the algorithmic principles of the learning framework are shown in the form of pseudocode.

```
Algorithm 1 CEEMDAN-TCN-ESN
Inputs: Electric load data
Outputs: Point estimation of short-term load forecasting
#Step 1:Modal decomposition
Initialize the parameters of CEEMDAN
Apply CEEMDAN algorithm to decompose the original electric load data
into n subseries according to Eqs(1)-(8).
#Step 2:Sequence reconstruction
Reconstructing the subseries obtained in Step 1 (except the trend
item) according to Eqs(9)-(11) to obtain the Higher-frequency series
and Lower-frequency series, and then the trend item is added to the
Lower-frequency series.
#Step 3:Model Selection
Initialize the parameters of all model
for i = 1:m(m is the total number of models used in the experiment)
Use Model i to fit and predict Higher-frequency series and obtain cor-
responding evaluation indicators.
Select the most appropriate model to predict Higher-frequency series
according to evaluation indicators.
for i = 1:m
Use Model i to fit and predict Lower-frequency-trend series and obtain
corresponding evaluation indicators.
Select the most appropriate model to predict Lower-frequency-trend
series according to evaluation indicators.
#Step 4:Forecast result integration
Add the prediction results of the two series in step 3
Return: Final predicted value and evaluation indicators
```

The learning framework of CEEMDAN-TCN-ESN is shown in Fig 5.

The CEEMDAN-TCN-ESN learning framework can be specifically divided into four steps.

1. Modal Decomposition.
   In this paper, first, after completing basic operations such as data preprocessing and smoothing for a time series, CEEMDAN is used to perform modal decomposition of the series to obtain *n* modal components.

2. Sequence Reconstruction.
   The modal components obtained in (1) are then reconstructed serially by summing the sub-series obtained from the CEEMDAN decomposition item by item, defining the sum of the first *i* IMFs as $index_i$ (*i = 1,2…. … n*), calculating the mean of $index_1$ to $index_n$, and conducting Student's t test on whether the mean value is significantly different from zero or not. At this point, the first *k-1* signal components are the higher-frequency components; thus, the first *k-1* modal components are combined to obtain the higher-frequency series, and the remaining components except the last component (trend term) are combined to obtain the lower-frequency series. The lower-frequency-trend series is obtained by merging the lower-frequency series with the trend term, which completes the reconstruction task of the series.

3. Model Selection.
   The higher-frequency and lower-frequency-trend series obtained from the reconstruction in (2) are fitted and predicted by various models, and the optimal prediction models for the two series are selected based on the previously determined model performance indicators (e.g., RMSE and MAE). The prediction data of the training set and the test set are obtained for the two series when the optimal models are applied for prediction.

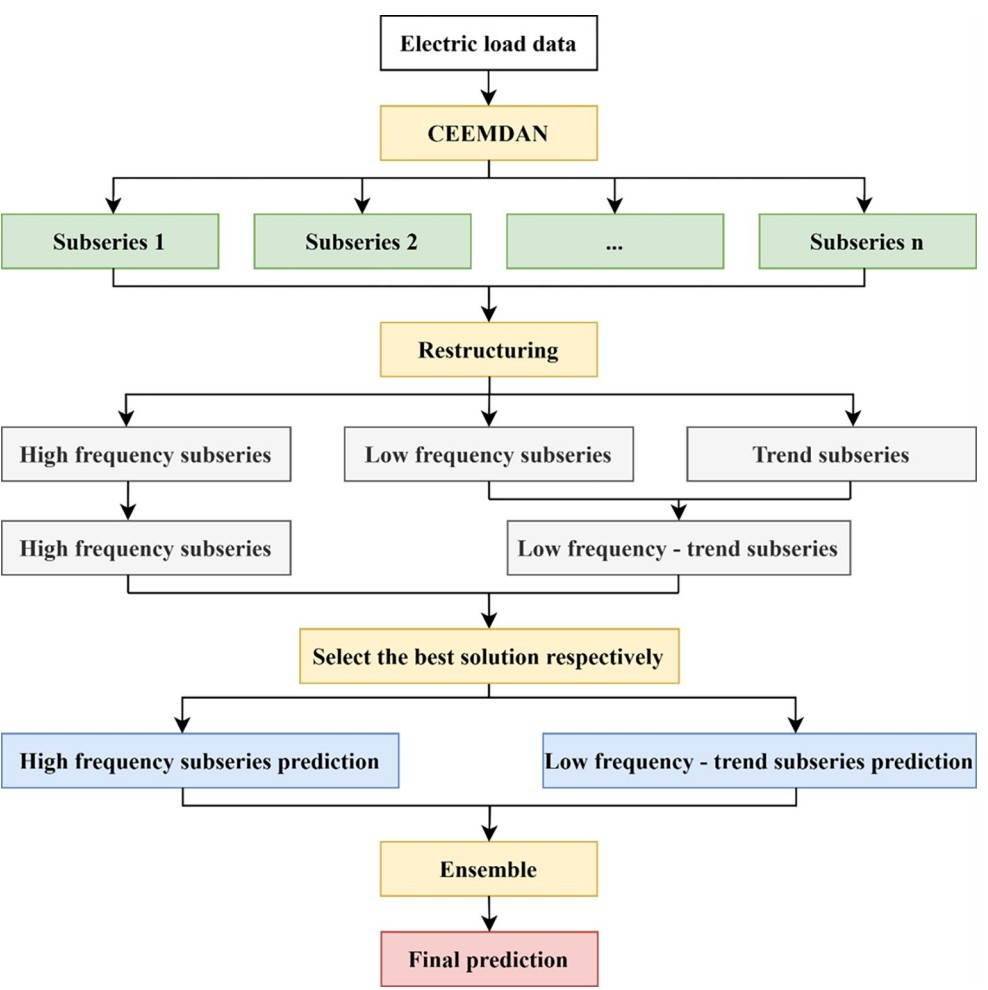

**Fig 5. The learning framework of CEEMDAN-TCN-ESN.**

4. Integration of Predicted Results.

The training-set prediction data of the two sequences in (3) are summed to obtain the final training-set prediction results and evaluation indicators of CEEMDAN-TCN-ESN; similarly, the test-set prediction data of the two sequences are summed to obtain the final test-set prediction results and evaluation indicators of CEEMDAN-TCN-ESN in this paper.

## 4 Experimental analysis

In this section, information about the dataset used, the indicators used to evaluate the predictive performance of the models, and the parameter settings of the various models chosen for the experiments are provided.

### 4.1 Data source

The data used in this paper are derived from the Panama national electricity load data. This dataset provides data on Panama's national electricity load at one-hour intervals and provides a total of 17,520 data points. The time span of the series is from 0:00 on January 1st, 2018, to 23:00 on December 31st, 2019. The statistical results of the raw data are shown in Table 1.

**Table 1. The statistical results of raw data.**

| Variable name | Sample size | Mean value | Standard deviation | Minimum value | Maximum Value |
|---|---|---|---|---|---|
| nat_demand | 17520 | 1224.629 | 188.256 | 85.193 | 1721.516 |

## 4.2 Evaluation indicators

To assess the prediction effectiveness of each model from a quantitative perspective, this paper chooses the root mean square error (RMSE), mean absolute error (MAE) and goodness-of-fit ($R^2$) to measure the prediction accuracy and generalization ability of different models. Assuming $y_i$ is the true data value and $\hat{y}_i$ is the predicted value of the model, where $i = 1,2...n$ ($n$ is the sample size), the mathematical representation of the abovementioned evaluation indicators is shown in Eqs (18)-(20).

$$RMSE = \sqrt{\frac{1}{n}\sum_{i=1}^{n}(y_i - \hat{y}_i)^2} \tag{18}$$

$$MAE = \frac{1}{n}\sum_{i=1}^{n}|y_i - \hat{y}_i| \tag{19}$$

$$R^2 = 1 - \frac{\sum_{i=1}^{n}(y_i - \hat{y}_i)^2}{\sum_{i=1}^{n}(y_i - \overline{y_i})^2} \tag{20}$$

From the definitions of the above three evaluation indicators, it can be seen that if the values of RMSE and MAE are smaller and the value of $R^2$ is closer to 1, the prediction error of the model will be smaller and the prediction effect of the model will be better, suggesting that the model's generalization ability may be stronger.

## 4.3 Experimental setup

1. Data preprocessing
   The current study uses power load data of two years in length and hourly in frequency, and the dataset does not contain missing values. Outliers in short-term power load forecasting have a large negative impact on the establishment of the model, so the identification and processing of outliers is critical in the training process. To achieve convenient and rigorous identification of outlier points, it is assumed that the electricity load data in the dataset obey a normal distribution. Then, according to the 3δ principle of a normal distribution, the electricity load values that differ from the sample mean by more than 3 times the standard deviation are identified as outliers, whereby five outlier points are detected in the electricity load data of Panama for two years (2018–2019) and replaced with missing values. Since the electric load data are continuous and periodic to some extent, this paper chooses the linear interpolation method to fill the above missing values. In addition, to avoid the problem of increasing the training time or even failing to converge due to the existence of odd sample data in the dataset, the data are normalized before training.

In this paper, the first 80% of the data in the dataset are used as the training set, and the remaining 20% are divided into the test set. The training set is used to select key parameters for the training model and to build the model, while the test set is used to evaluate the prediction accuracy of the model.

2. Single prediction model parameter settings

   In this experiment, eight single models are selected to model the time series directly after data preprocessing. The eight single models selected are ARIMA, LSTM, BILSTM, CNN, TCN, GRU, RBF, and ESN. The parameters of each single model are set as follows.

   ARIMA: First, the series is tested for smoothness and white noise, and the test results show that the series is a smooth and nonwhite noise series after data preprocessing. Then, the autoregressive order, moving average order and differential order of the ARIMA model are set as p, q and d, respectively, and the ranges of p, q and d are set at (0, 2). Combining the AIC and SBC, the parameters of the ARIMA model are finally determined as follows: p = 2, q = 1, and d = 1, which means fitting an ARIMA(2,1,1) model.

   BILSTM: The model parameters of BILSTM in this test are set as follows. The number of hidden layer and hidden layer units is set to 2 and 32, respectively, the learning rate is set to 0.01, and the training round number is set to 80. "Adam" is chosen as the optimizer, the loss function is MSE, and the batch size is 32.

   LSTM: The model parameters of LSTM in this test are set as follows. "ReLU" is chosen as the activation function, the learning rate and the discard rate are set to 0.01 and 0.2, respectively, and the training round number is set to 80. "Adam" is chosen as the optimizer, the loss function is MSE, and the batch size is 32.

   CNN: The model parameters of the CNN in this test are set as follows. The number of filters and the convolutional kernel size are set to 32 and 2, respectively, "ReLU" is chosen as the activation function, both the number of strides and pool_size are set to 1, the training round number is set to 80, "Adam" is chosen as the optimizer, the loss function is MSE and the batch size is 16.

   GRU: The model parameters of the GRU in this test are set as follows. The number of hidden layer units and the discard rate are set to 100 and 0.2, respectively, the output layer contains 1 neuron, and the number of training rounds is set to 80. "Adam" is chosen as the optimizer, the loss function is MSE, and the batch size is 32.

   RBF: The model parameters of RBF in this test are set as follows. The radial basis function expansion rate is set to 1000.

   The parameters of the TCN and ESN are specified in (5) and (6).

3. CEEMDAN parameter settings

   In this paper, the CEEMDAN algorithm is used in the PyEMD package to set different modal numbers to decompose and test the training set data. The results show that when the modal number is set to 15, the score of each subseries obtained after CEEMDAN decomposition is the most stable.

4. Sequence reconstruction parameter setting

   This paper uses MATLAB to reconstruct all the subseries obtained after CEEMDAN decomposition except the trend term according to the high- and low-frequency characteristics and define the sum of the first i signal components as $index_i$. The basis for dividing the higher-frequency series and lower-frequency series is set as whether the mean value of the first k indexes is significantly different from zero or not. If it is not significantly different from zero, the first k signal components will be divided into a higher-frequency series, and if it is significantly different from zero, then the first k-1 components will be classified as a higher-frequency series, and the remaining components will constitute a lower-frequency-trend series with a trend term.

5. TCN parameter settings

**Table 2. Key parameters of the TCN.**

| Main hyperparameter | Set value |
| --- | --- |
| Nb Filters | 64 |
| Kernel Size | 3 |
| Nb stacks | 1 |
| Dilations | [1,2,4,8] |
| Padding | Causal |
| Dropout rate | 0.2 |
| Activation | ReLU |
| Batch size | 32 |
| Max epoch | 80 |
| Optimizer | Adam |
| Loss function | MSE |

In this paper, the higher-frequency series obtained after reconstruction are used as the input series of the TCN prediction model. Based on the theoretical research in related aspects and combined with the specific dataset selected in this paper, the key parameters of the TCN are finally shown in Table 2.

Since the trend of power load data tends to be temporally periodic, and generally on a daily basis, the input of each sample of the TCN network in this paper is the power load data of the previous 24 hours, and the output is the point estimate of the current power load forecast.

The TCN network consists of one residual module, followed by a fully connected layer, in which the residual module contains two convolutional units and one nonlinear mapping. To extract the sample information more fully, the convolutional kernel size of the convolutional unit is preset to the common Figs 2 and 3, and it is found that the prediction effect is significantly better when the convolutional kernel size is 3 than 2 after preliminary experiments. The dropout layer is established to prevent overfitting, and the dropout rate is set to 0.2 (i.e., some neurons will be randomly deleted with a probability of 0.2). The expansion coefficient (dilations) is set to [1, 2, 4, 8] (exponentially increasing by 2), and the number of filters is set to 64. Since temporal signal modeling cannot violate the temporal order, which will produce causal convolution, the padding parameter is set as 'Causal'. For the choice of model optimizer, the Adam optimizer has a faster and more stable convergence speed in this experiment compared to stochastic gradient descent. For other parameters, the number of batch sizes and max epochs are set to 32 and 80, respectively, and the loss function is chosen as the commonly used MSE.

(2) ESN parameter settings: The lower-frequency series obtained after reconstruction are used as the input series of the ESN prediction model. Based on the theoretical research in related aspects and combined with the specific dataset selected in this paper, the key parameters of ESN are finally shown in Table 3.

For an echo state network (ESN), the main parameters include the size of the reserve pool ($N_R$), the spectral radius ($\rho$), the input scale factor ($W_{in}$), and the output regularization factor ($\lambda_r$). MATLAB is used to complete the ESN prediction experiment for lower-frequency-trend series, setting the number of samples in the initialized reserve pool to 100, the size of the reserve pool to 800, the update rate to 0.001, and the output regularization factor to 0.001. During the propagation of the echo state network, the weight matrix from the input layer to the reserve pool ($W_{in}$), the weight matrix of the output layer to the reserve pool ($W_{back}$), and the

**Table 3. Key parameters of the ESN.**

| Main hyperparameter | Set value |
|---|---|
| Initialize | 100 |
| Hidden | 800 |
| Learning Rate | 0.001 |
| Regularization coefficient | 0.001 |
| Optimizer | Adam |
| Loss function | MSE |

internal neuron connection weight matrix ($W$) in the reserve pool are set in advance and generated randomly. The spectral radius is the maximum value in the mode of all eigenvalues of the weight matrix $W$. For the selection of the optimizer, "Adam" is chosen. The loss function is still the commonly used MSE.

## 4.4 Experimental results and analysis

**4.4.1 Single model prediction.** To demonstrate the enhancement effect of CEEMDAN decomposition on short-term electricity load forecasting, this paper first uses different single models for short-term electricity load forecasting without decomposing the sequence, and eight different forecasting methods (BiLSTM, ARIMA, GRU, RBF, CNN, LSTM, TCN, and ESN) are selected for experiments. The evaluation results of each model on the test set (Panama national electricity load data from 4:00 on August 8th, 2019, to 23:00 on December 31st, 2019) are shown in Table 4 and Fig 6.

After analyzing the results of the single model forecasting experiments and considering the three evaluation indicators, this paper finds that the TCN model has the best forecasting effect on this dataset, BILSTM is the second best single forecasting model, and the ARIMA model has the worst forecasting effect. This shows that the two neural network models, TCN and BILSTM, have excellent performance in the short-term electric load forecasting task. The possible reasons for this are as follows: (1) short-term power load data are often influenced by many factors, and they often have complex and variable characteristics with weak regularity, so using traditional statistical models (e.g., ARIMA) to forecast them can only extract linear trends or some simple features in the data, so the prediction results will be worse. Compared with the traditional statistical model, the neural network model has more advantages in processing data with complex and changeable characteristics. (2) For the short-term power load prediction problem, RNNs (such as LSTM and BILSTM) may work better than RBF and CNN, and LSTM has an extra gate compared with GRU, which can be used to control the flow of information, so it may improve the prediction effect. (3) Meanwhile, BILSTM, as a two-way

**Table 4. Evaluation results of single prediction model indicators.**

| Models | RMSE | MAE | $R^2$ |
|---|---|---|---|
| ARIMA | 43.056 | 31.154 | 0.948 |
| CNN | 41.513 | 32.157 | 0.952 |
| GRU | 36.252 | 27.782 | 0.963 |
| RBF | 34.972 | 26.099 | 0.966 |
| ESN | 34.994 | 26.131 | 0.966 |
| LSTM | 33.553 | 25.459 | 0.968 |
| BILSTM | 32.521 | 24.684 | 0.970 |
| **TCN** | **30.965** | **23.209** | **0.973** |

(a)    (b)

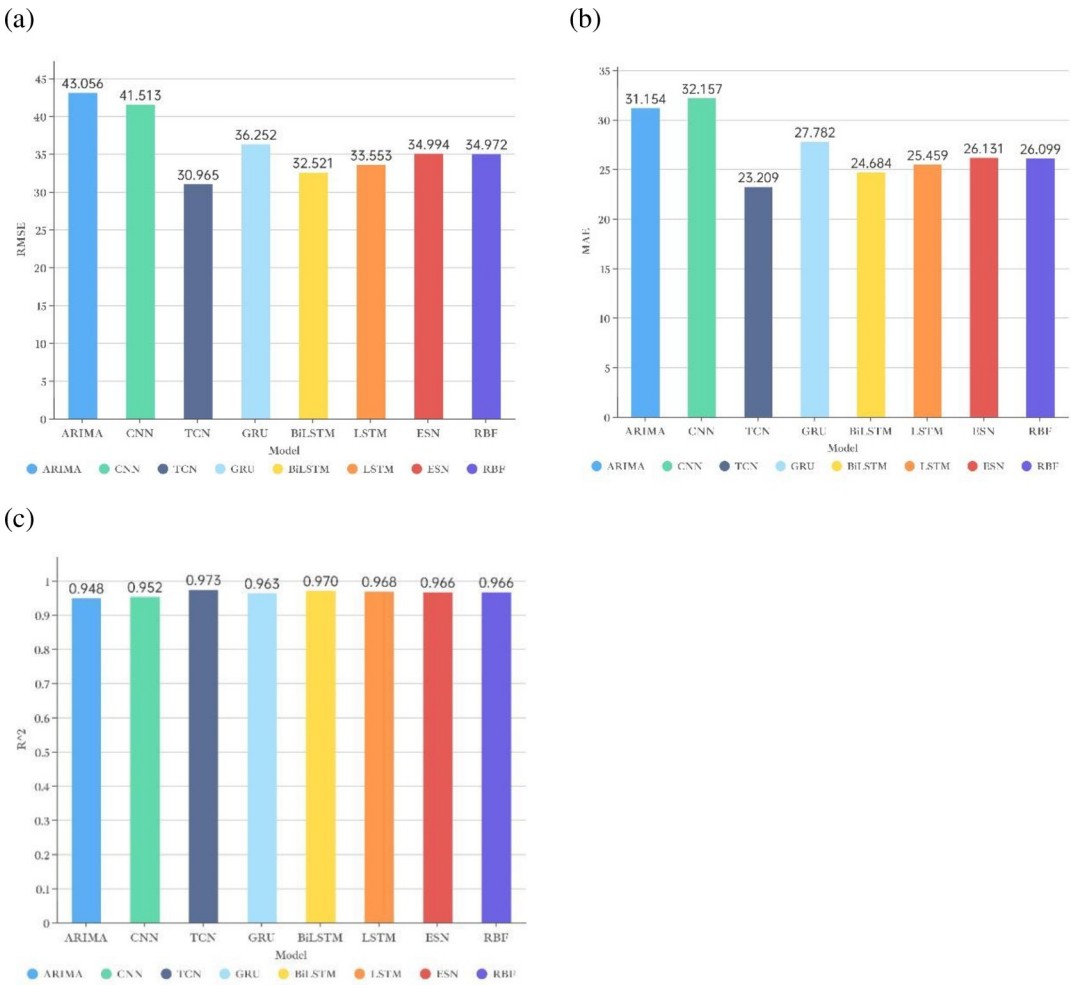

(c)

**Fig 6.** Comparative bar chart of single model prediction evaluation indicators: (a) RMSE indicator; (b) MAE indicator; (c) $R^2$ Indicators.

(long-term and short-term memory), consists of a combination of forward LSTM and backward LSTM, which can better capture the relationship between the sequence of temporal features than the unidirectional LSTM, taking into account both the information in front and behind. Since time series prediction sometimes needs to use both previous and later data as input items to build the model, BILSTM can improve the prediction effect compared with LSTM. (4) Unlike LSTM, which can only process time series data sequentially, TCN can process time series data in parallel. At the same time, the TCN contains many parameters and has a more complex structure. TCN has a flexible perceptual field (jointly determined by the size of the convolution kernel, the dilation coefficient, and the number of residual blocks), so it can flexibly cope with different prediction problems by adjusting the parameters. In addition, LSTM often suffers from gradient disappearance and gradient explosion, while TCN circumvents these problems.

In summary, TCN performs best when the original series is predicted directly using a single model after data preprocessing only. Considering the complexity of short-term power load data features, this paper next performs a modal decomposition of the series to fully extract the features of this series and complete further prediction tasks.

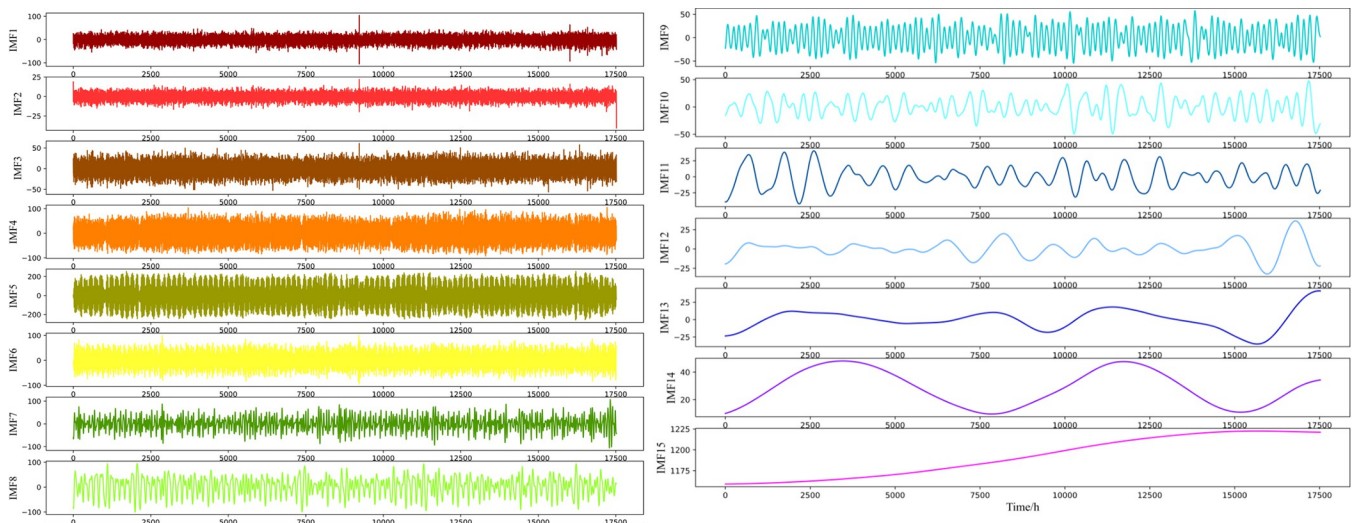

**Fig 7. CEEMDAN decomposition result.**

**4.4.2 CEEMDAN modal decomposition.** In this paper, according to the CEEMDAN modal decomposition method, the original Panamanian national electricity load data series is decomposed into 15 sets of decomposition waves, as shown in Fig 7.

The original Panamanian national electricity load data series is decomposed into several IMF components with different frequencies, and the IMF component series with lower frequencies show a smoother trend. The original time series is divided into 15 subseries after CEEMDAN decomposition. The frequency of the signal components from IMF1 to IMF15 gradually decreases, the amplitude gradually decreases, the wavelength gradually increases, and there is no obvious trend of change in each subseries. The smoothness of the decomposed subseries increases compared with the original series, and the noise contained in the sequence gradually decreases; thus, sequence decomposition is likely to improve the prediction effect.

**4.4.3 IMF component reconstruction results.** Since the original Panamanian national electric load data series will obtain many decomposition modes after decomposition, the decomposition modes with high similarity need to be reconstructed to avoid many feature calculations and manual analysis. Considering that different models have significant differences in applicability in both high- and low-frequency data, while the characteristics between the series attributed to the same frequency are approximately the same, i.e., the models have basically the same prediction effect on them, higher-frequency, lower-frequency and trend components are selected for discriminant reconstruction. This paper first sums the IMF components obtained from CEEMDAN decomposition item by item, defines the sum of the first $i$ IMFs as $index_i$ ($i = 1,2....k$), calculates the mean value of $index_1$ to $index_k$, and performs Student's t test to determine whether the mean value is significantly different from zero to discriminate the high- and low-frequency components. The discriminant result is that the first four signal components, namely, IMF1-IMF4, are high-frequency components, IMF5-IMF14 are low-frequency components, and IMF15 is the trend term. The higher-frequency and lower-frequency components are then summed separately to achieve sequence reconstruction. The reconstructed higher-frequency, lower-frequency and trend (partial) series are shown in Fig 8.

**4.4.4 TCN experimental results.** To compare the prediction performance of different models in terms of time series of higher-frequency electricity load data after reconstructing the IMF components, eight different prediction methods such as BiLSTM, ARIMA, GRU, RBF,

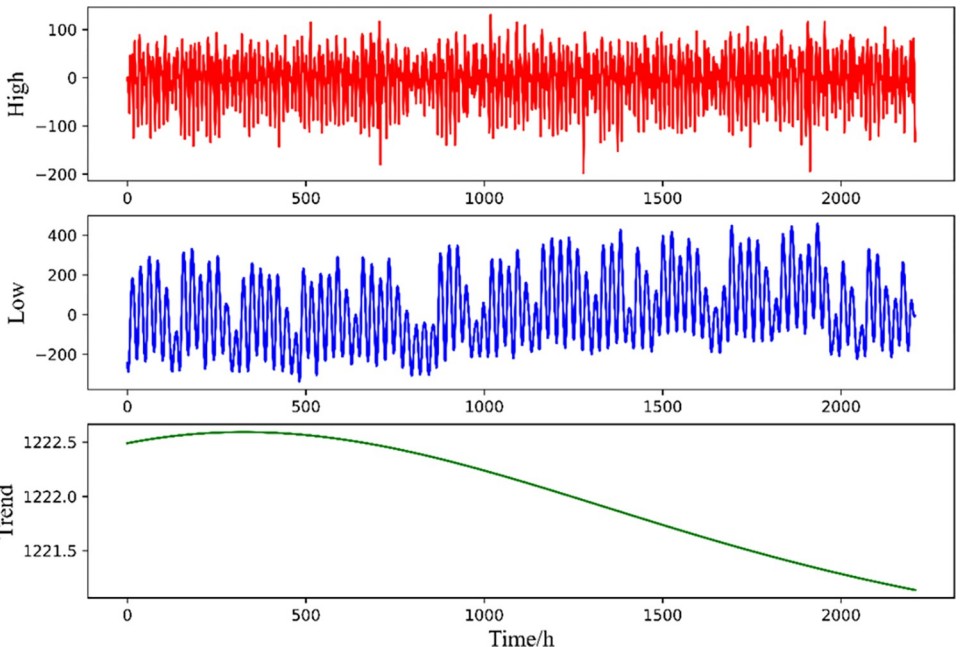

**Fig 8. IMF component reconstruction results.**

CNN, LSTM, TCN, and ESN were selected for experiments in this paper. The evaluation results and prediction results of each model on the test set (Panama national electricity load data from 4:00 on August 8th, 2019, to 23:00 on December 31st, 2019) are shown in Table 5 and Fig 9.

Based on the graphs, the evaluation indicators of the TCN model are found to be optimal compared to the evaluation indicators of other models after analysis, which indicates that the TCN model has excellent performance in the prediction of higher-frequency power load data. The prediction process of the TCN model for higher-frequency electric load data is shown in Fig 10.

This paper analyzes the possible reasons for the excellent performance of the TCN model compared to other models in predicting higher-frequency electric load data as follows: (1) First, TCN can perform parallel processing when making predictions while facing time series sensitive problem tasks, although RNN (e.g., LSTM) may usually be more suitable than BP and RBF. It can be used to control the information flow and thus may improve the prediction results, but compared with TCN, it can only perform sequential processing. Therefore, it may

**Table 5. Evaluation results of each prediction model for high-frequency data indicators.**

| Models | RMSE | MAE | $R^2$ |
|---|---|---|---|
| ARIMA | 37.822 | 27.762 | 0.422 |
| CNN | 19.601 | 14.366 | 0.845 |
| **TCN** | **14.971** | **10.805** | **0.909** |
| LSTM | 17.898 | 13.187 | 0.871 |
| BiLSTM | 17.355 | 12.774 | 0.878 |
| GRU | 17.936 | 13.360 | 0.870 |
| ESN | 20.542 | 15.347 | 0.829 |
| RBF | 20.522 | 15.339 | 0.830 |

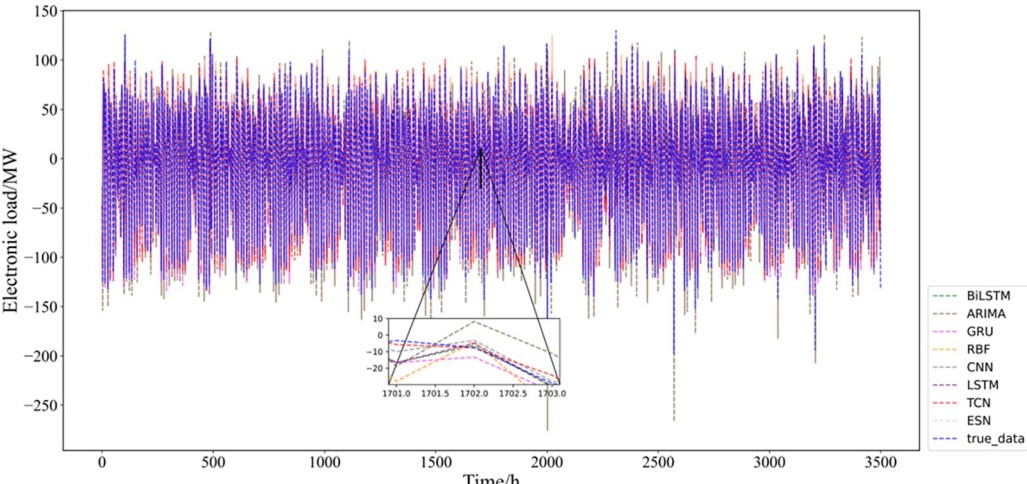

**Fig 9. Experimental results of the prediction of high-frequency data by each model.**

be less effective than TCN for higher-frequency time series prediction. (2) Second, TCN has a flexible perceptual field, which is determined by the number of layers, convolutional kernel size, and expansion coefficient. It can be flexibly customized according to the different characteristics of different tasks. Therefore, when dealing with high-frequency data, there is no problem of gradient disappearance and gradient explosion, as other models often have. Some research results show that TCNs with the introduction of architectural elements such as null convolution and residual connectivity are more effective than recursive architectures such as LSTM in different time series modeling tasks. (3) While ARIMA is a traditional time series forecasting method, its accuracy is relatively poor compared to that of neural network forecasting models. Moreover, ESN presents crossover prediction results among the eight models, presumably because ESN is not suitable for time series prediction modeling of this dataset.

Therefore, TCN is improved in different aspects compared with other models, so the TCN model is chosen to predict the time series of high-frequency power load data after the reconstruction of IMF components.

**4.4.5 ESN experimental results.** To compare the prediction performance of different models in terms of time series of electricity load data with low frequency and trend terms after reconstructing the IMF components, 8 different prediction methods (BiLSTM, ARIMA, GRU, RBF, CNN, LSTM, TCN, and ESN) were selected for experiments. The evaluation results and prediction results of each model on the test set are shown in Table 6 and Fig 11.

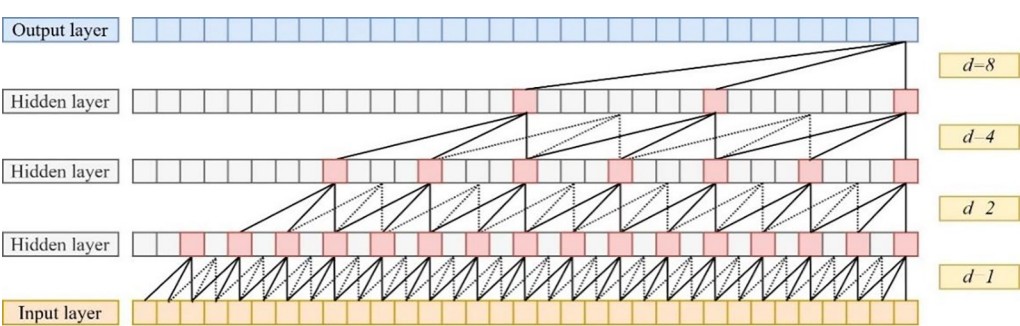

**Fig 10. The prediction process of the TCN model for higher-frequency electric load data.**

Table 6. Evaluation results of each prediction model for low frequency plus trend terms data indicators.

| Models | RMSE | MAE | $R^2$ |
|---|---|---|---|
| ARIMA | 2.454 | 1.973 | 0.999 |
| CNN | 20.117 | 16.936 | 0.986 |
| TCN | 8.219 | 6.705 | 0.997 |
| LSTM | 12.790 | 10.377 | 0.994 |
| BiLSTM | 9.187 | 7.377 | 0.997 |
| GRU | 9.298 | 7.679 | 0.997 |
| **ESN** | **1.582** | **1.255** | **0.999** |
| RBF | 1.999 | 1.578 | 0.999 |

Based on the graphs, the evaluation indicators of the ESN model are found to be optimal compared to the evaluation indicators of other models after analysis, which indicates that the ESN model has excellent performance in predicting low frequency and trend term power load data.

This paper analyzes the possible reasons for the excellent performance of the ESN model compared with other models as follows: (1) Compared with traditional recurrent neural networks, the biggest advantage of ESN is that it simplifies the training process of the network, solves the problems of the traditional recurrent neural network structure that is difficult to determine and overly complex training algorithms, and overcomes the problems of memory reduction in recursive networks. (2) Compared with CNN, LSTM and other neural network models with more complex structures, ESN uses a large-scale random sparse network (reserve pool) as the information processing medium, maps the input signal from the low-dimensional input space to the high-dimensional state space, and uses a linear regression method to train some connection rights of the network in the high-dimensional state space, while the other connection rights are randomly generated and kept constant during the network training process. It also effectively avoids the problem of overfitting.

Therefore, according to the above analysis, the ESN is improved in different aspects compared with other models, so the ESN model is chosen in this paper to predict the time series of lower-frequency and trend series.

**4.4.6 CEEMDAN-TCN-ESN experimental results.** Based on the above analysis, the TCN model is chosen to predict the reconstructed higher-frequency series, while the ESN model is chosen to predict the reconstructed lower-frequency and trend series. The TCN model, which is the most effective among the single models, is then used to predict all 15 subseries obtained from the original series after CEEMDAN decomposition, and the evaluation indicators of the two methods are compared to highlight the superiority of the CEEMDAN-TCN-ESN model. That is, this paper compares the prediction performance of the two models resulting from whether the IMF components are reconstructed with respect to the time series of power load data and the improvement effect of the above two models with respect to the prediction effect of the best single prediction model TCN without decomposition. In this paper, based on the single model prediction in 4.4.1, 3 different prediction methods (CEEMDAN-ESN, CEEMDAN-TCN and CEEMDAN-TCN-ESN) are further selected for the experiments. The evaluation indicators and forecasting results of the 5 models on the test set are shown in Table 7 and Fig 12.

In this paper, the superiority of the CEEMDAN-TCN-ESN model is demonstrated based on the comparison model. Based on the results in Fig 12, the following conclusions can be drawn: comparing the evaluation indicators of the model without the reconstruction strategy with that of the model with the reconstruction strategy, the following conclusions can be

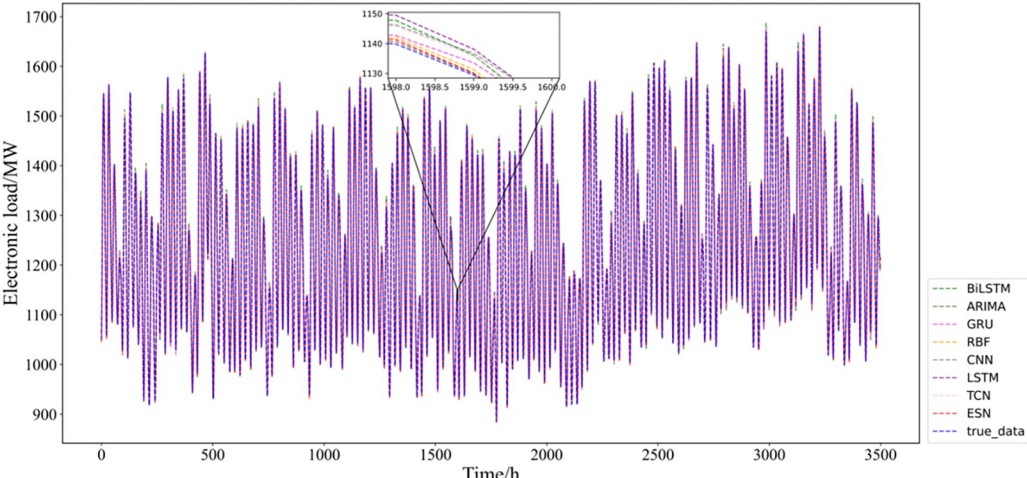

**Fig 11. Experimental results of the prediction of each model for low frequency plus trend term data.**

drawn: (1) The prediction effects of the CEEMDAN-TCN model and the CEEMDAN-ESN model with the CEEMDAN modal decomposition-based approach are significantly improved compared with the single TCN model and the ESN model, respectively. obtained a significant improvement, i.e., the CEEMDAN-TCN and CEEMDAN-ESN models reduced RMSE by 46.17% and 32.95% and MAE by 42.92% and 30.02%, respectively, compared to the TCN model and ESN model without the decomposition strategy. $R^2$ increased by 1.95% and 1.96%, respectively. This indicates that the high-frequency, low-frequency, and trend term subseries out of the CEEMDAN-based decomposition are smoother and more regular, which in turn reduces the effect of data instability on the prediction effect and indicates that TCN and ESN are more applicable to the prediction of the CEEMDAN-decomposed series. (2) The CEEM-DAN-TCN-ESN model with the reconstruction strategy is more suitable than the CEEM-DAN-TCN-ESN model without the reconstruction strategy. For the CEEMDAN-TCN model and CEEMDAN-ESN model, the RMSE decreased by 9.52% and 35.73%, and the MAE decreased by 17.39% and 40.15%, respectively. $R^2$ increased by 0.20% and 0.91%, respectively. It shows that the CEEMDAN-TCN-ESN model has a significant improvement in model evaluation indexes compared with the CEEMDAN-TCN model and the CEEMDAN-ESN model, indicating that the target model proposed in this paper has a higher prediction accuracy, better prediction performance, and a significant improvement in the prediction effect. This may be due to the division into two series of high-frequency and low-frequency trends in the reconstruction process. Then, relatively suitable models are selected for modeling and prediction according to the characteristics of different sequences to enhance the final prediction effect. Additionally, the summation of high-frequency, low-frequency and trend terms after their

**Table 7. Evaluation results of the forecast model on the indicators of electric load data.**

| Models | RMSE | MAE | $R^2$ |
|---|---|---|---|
| ESN | 34.994 | 26.131 | 0.966 |
| TCN | 30.965 | 23.209 | 0.973 |
| CEEMDAN-ESN | 23.465 | 18.287 | 0.985 |
| CEEMDAN-TCN | 16.667 | 13.247 | 0.992 |
| CEEMDAN-TCN-ESN | 15.081 | 10.944 | 0.994 |

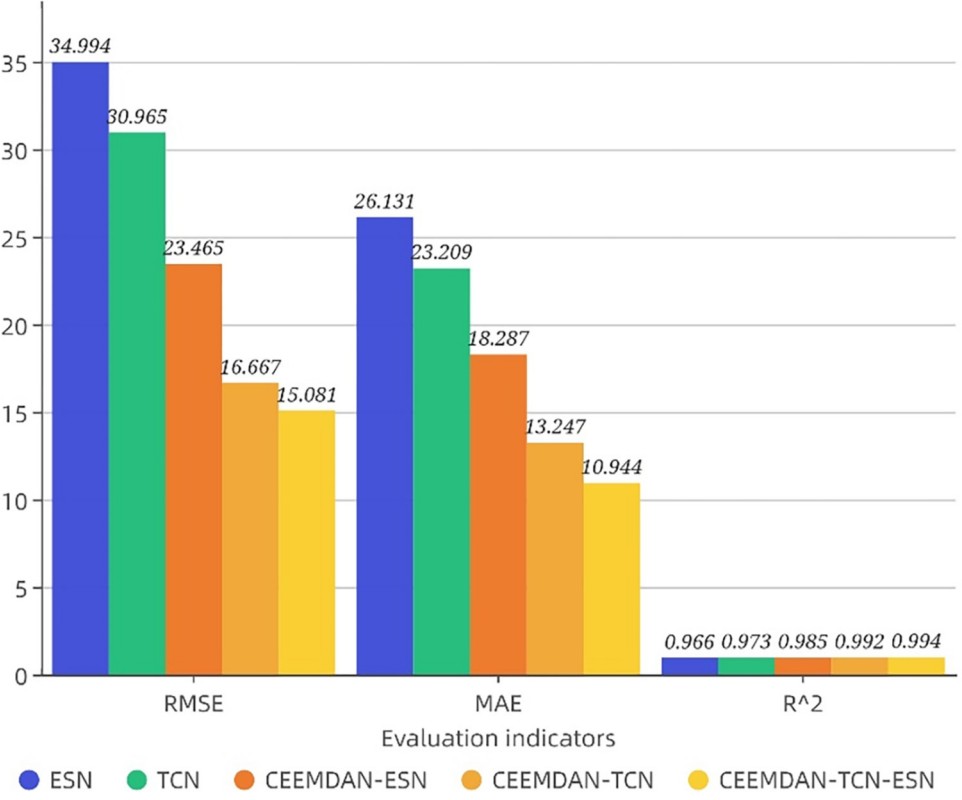

**Fig 12. Comparison bar chart of model prediction evaluation indicators.**

identification also has a certain degree of denoising effect, so the prediction effect is also improved.

## 5 Robustness analysis

Robustness analysis is critical and necessary for better application of the model to real-life applications. The data used in this section are derived from the solar power hourly data of the United States in the Kaggle data platform's short-term electricity load forecasting dataset. A total of 19445 solar power hourly data points (from 0:00 on January 1st, 2020 to 4:00 on March 21st, 2022) in 48 states of the United States are selected for the robustness analysis experiments of the proposed model. The first 16445 data points are used as the training set, and the last 3000 data points are used as the test set. The final prediction results of each model on the test set are shown in Table 8 and Fig 13.

**Table 8. Evaluation results of the forecast model on the indicators of solar power hourly data.**

| Models | RMSE | MAE | $R^2$ |
|---|---|---|---|
| ESN | 1518.755 | 994.678 | 0.985 |
| TCN | 1259.476 | 987.299 | 0.989 |
| CEEMDAN-ESN | 1121.479 | 872.435 | 0.992 |
| CEEMDAN-TCN | 1160.891 | 692.875 | 0.992 |
| CEEMDAN-TCN-ESN | 870.532 | 595.843 | 0.995 |

(a)

(b)

(c)

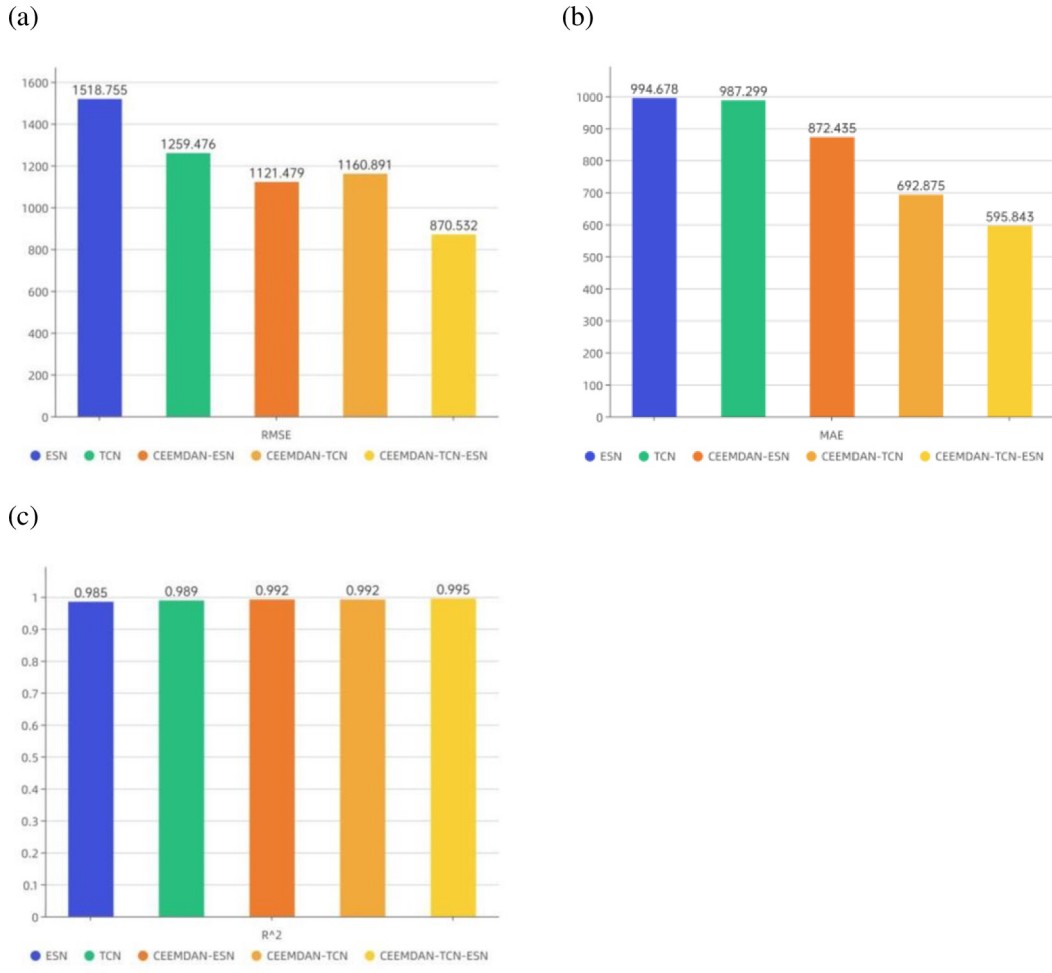

**Fig 13.** Comparative bar chart of model prediction evaluation indicators: (a) RMSE indicator; (b) MAE indicator; (c) $R^2$ Indicators.

From Table 8 and Fig 13, it is clear that the hybrid CEEMDAN-TCN-ESN model proposed in this paper still has good prediction results when applied to other datasets, which suggests that the model has strong robustness, and the model's adaptability to different datasets and prediction results are more satisfactory.

## 6 Conclusion

Due to the complex characteristics of the data and many related influencing factors, this paper establishes a hybrid model of the TCN and ESN based on CEEMDAN decomposition and high- and low-frequency series reconstruction modes for the STLF problem. In this paper, CEEMDAN decomposition is used to decompose the original power load series into subseries with different frequency characteristics, and all the subseries are reconstructed according to their frequency characteristics to obtain the higher-frequency and the lower-frequency-trend series. Then, the same eight models are used to predict the higher-frequency and lower-frequency-trend series, and the best prediction models for the two reconstructed series are TCN and ESN. Finally, the prediction results for the two series are combined to output the final power load forecast results.

Through the experimental analysis and the comparative analysis of different models, the results show the following: (1) For the prediction of the original electricity load series, TCN has the best performance among the single prediction models chosen for this experiment, indicating that it can extract the features of the original time series to a large extent. BILSTM ranks second, and ARIMA is the worst, indicating that deep learning approaches generally outperform traditional statistical models in STLF. (2) CEEMDAN decomposition can fully extract time series features and significantly improve the prediction accuracy of the model. (3) Sequence reconstruction can fuse the sequences with similar frequency features in the subseries obtained from CEEMDAN decomposition to obtain two reconstructed series. The optimal model is selected for each of the two series to produce a hybrid model to further improve the prediction effect of the model. (4) The RMSE, MAE, and $R^2$ of the proposed hybrid model are 15.081, 10.944, and 0.994, respectively, which are higher than those of the other models. Compared to the best single prediction model (TCN), the RMSE is reduced by 51.3%, the MAE is reduced by 52.85%, and the $R^2$ is improved by 2.16%. Compared to the second-best model (CEEMDAN-TCN), the RMSE is reduced by 9.52%, the MAE is reduced by 17.39%, and the $R^2$ is improved by 0.20%.

The CEEMDAN-TCN-ESN model proposed in this paper performs well in STLF and has certain application prospects, but the model still has the following problems: the expenditure of the model is large, the depth of the TCN is shallow, and only the hour-by-hour forecasting of short-term power load data is performed. In the future, if the experimental budget is sufficient, deepening the depth of the network to explore the prediction accuracy of the proposed model for 3 h, 6 h, 12 h or even 24 h in the future and expanding the scope of the model could be considered.

## Author Contributions

**Conceptualization:** Jiacheng Huang, Xuchu Jiang.

**Data curation:** Jiacheng Huang, Xiaowen Zhang, Xuchu Jiang.

**Formal analysis:** Xiaowen Zhang, Xuchu Jiang.

**Funding acquisition:** Xuchu Jiang.

**Investigation:** Xiaowen Zhang, Xuchu Jiang.

**Methodology:** Xiaowen Zhang.

**Writing – original draft:** Jiacheng Huang.

**Writing – review & editing:** Xuchu Jiang.

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
