## [Decision Letter · Decision Letter 0]

10 Feb 2023

PONE-D-23-03222Short-Term Power Load Forecasting Based on the CEEMDAN-TCN-ESN ModelPLOS ONE

Dear Dr. Jiang,

Thank you for submitting your manuscript to PLOS ONE. After careful consideration, we feel that it has merit but does not fully meet PLOS ONE’s publication criteria as it currently stands. Therefore, we invite you to submit a revised version of the manuscript that addresses the points raised during the review process.

We look forward to receiving your revised manuscript.

Kind regards,

Lin Wang, Ph.D.

Academic Editor

PLOS ONE

Journal Requirements:

2.  Our internal editors have looked over your manuscript and determined that it is within the scope of our Smart Energy Systems Call for Papers. The Collection will encompass the latest research in smart grid technologies, including information technologies, device integration, distribution methods, and data mining, all towards improving the efficiency of energy supply networks. Additional information can be found on our announcement page: https://collections.plos.org/call-for-papers/smart-energy-systems/. If you would like your manuscript to be considered for this collection, please let us know in your cover letter and we will ensure that your paper is treated as if you were responding to this call. If you would prefer to remove your manuscript from collection consideration, please specify this in the cover letter.

4. We note that Figure 4 in your submission contain map images which may be copyrighted. All PLOS content is published under the Creative Commons Attribution License (CC BY 4.0), which means that the manuscript, images, and Supporting Information files will be freely available online, and any third party is permitted to access, download, copy, distribute, and use these materials in any way, even commercially, with proper attribution. For these reasons, we cannot publish previously copyrighted maps or satellite images created using proprietary data, such as Google software (Google Maps, Street View, and Earth). For more information, see our copyright guidelines: http://journals.plos.org/plosone/s/licenses-and-copyright.

a. You may seek permission from the original copyright holder of Figure 4 to publish the content specifically under the CC BY 4.0 license.  

Additional Editor Comments:

Please consider the reviewers' suggestions and give a point-to-point answer.

Reviewers' comments:

Reviewer's Responses to Questions

**Comments to the Author**

1. Is the manuscript technically sound, and do the data support the conclusions?

Reviewer #1: Yes

Reviewer #2: Yes

2. Has the statistical analysis been performed appropriately and rigorously? 

Reviewer #1: Yes

Reviewer #2: Yes

3. Have the authors made all data underlying the findings in their manuscript fully available?

Reviewer #1: Yes

Reviewer #2: Yes

4. Is the manuscript presented in an intelligible fashion and written in standard English?

Reviewer #1: Yes

Reviewer #2: Yes

5. Review Comments to the Author

Reviewer #1: In this study, a hybrid CEEMDAN-TCN-ESN forecasting model is proposed for short-term load forecasting research. The works done in the paper are good, but to make it acceptable, some improvements must be made.

1. Please check Eqs. (13)-(15), What is in the brackets?

2. Some raw data should be displayed. The statistical results of raw data should be presented.

3. MAPE can be added to show the performance.

4. As written, some imprecision in word choice, grammar, and punctuation from the conveyance of the research.

5. The literature review should be improved since many advanced models (machine learning, deep learning, and so on) have been proposed in recent years. Their merits and demerits should be discussed to enrich this study. Some of the examples are:

[1] https://doi.org/10.1016/j.energy.2021.121756

[2] https://doi.org/10.1016/j.dsm.2021.12.002

[3] https://doi.org/10.1016/j.dsm.2022.09.001

Reviewer #2: The topic is important. However, the following problems must be handled.

(a) How to set the related parameters of the proposed CEEMDAN-TCN-ESN for better performance?

(b)The Conclusions have not been written well. It is better to explain briefly the research gaps, research objectives, and methodology, and then clearly present the main results obtained in the work.

(c)Section 1.4: Some advances of deep learning methods with necessary analysis can be included to improve the importance (Multivariate wind speed forecasting based on multi-objective feature selection approach and hybrid deep learning model; Deep learning combined wind speed forecasting with hybrid time series decomposition and multi-objective parameter optimization; etc.).

(d)Reference: Some old papers should be deleted. Plz add new papers since 2021.

(e) Plz further highlight the contribution of this study.

6. PLOS authors have the option to publish the peer review history of their article (what does this mean?). If published, this will include your full peer review and any attached files.

Reviewer #1: No

Reviewer #2: No

---

## [Author Response · Author response to Decision Letter 0]

15 Feb 2023

Thank you very much for your comments. These opinions are very helpful and enlightening for our article. Our response to the comments is uploaded to the submission system. Please check.

---

## [Decision Letter · Decision Letter 1]

23 Mar 2023

PONE-D-23-03222R1Short-Term Power Load Forecasting Based on the CEEMDAN-TCN-ESN ModelPLOS ONE

Dear Dr. Jiang,

Thank you for submitting your manuscript to PLOS ONE. After careful consideration, we feel that it has merit but does not fully meet PLOS ONE’s publication criteria as it currently stands. Therefore, we invite you to submit a revised version of the manuscript that addresses the points raised during the review process.

We kindly ask you to revise the paper considering the Reviewers' remarks and suggestions presented below. When this process is completed, the paper may be acceptable for publication.==============================

We look forward to receiving your revised manuscript.

Kind regards,

Lin Wang, Ph.D.

Academic Editor

PLOS ONE

Reviewers' comments:

Reviewer's Responses to Questions

**Comments to the Author**

1. If the authors have adequately addressed your comments raised in a previous round of review and you feel that this manuscript is now acceptable for publication, you may indicate that here to bypass the “Comments to the Author” section, enter your conflict of interest statement in the “Confidential to Editor” section, and submit your "Accept" recommendation.

Reviewer #1: All comments have been addressed

Reviewer #2: All comments have been addressed

2. Is the manuscript technically sound, and do the data support the conclusions?

Reviewer #1: Yes

Reviewer #2: Partly

3. Has the statistical analysis been performed appropriately and rigorously? 

Reviewer #1: Yes

Reviewer #2: I Don't Know

4. Have the authors made all data underlying the findings in their manuscript fully available?

Reviewer #1: No

Reviewer #2: Yes

5. Is the manuscript presented in an intelligible fashion and written in standard English?

Reviewer #1: Yes

Reviewer #2: Yes

6. Review Comments to the Author

Reviewer #1: (No Response)

Reviewer #2: This revision is improved. For the technology aspect, the contribution is not very strong. However, some of my advices such as the possible applications of deep learning methods in this field are not handled. The authors just discussed it in the response letter. It is better to analyzed in the further research with some references. The writing should be improved.

The contributions are given. But some are not convincing. 3-4 Points are enough.

7. PLOS authors have the option to publish the peer review history of their article (what does this mean?). If published, this will include your full peer review and any attached files.

Reviewer #1: No

Reviewer #2: No

---

## [Author Response · Author response to Decision Letter 1]

24 Mar 2023

Thank you for the reviewer's comments concerning our manuscript. These comments are all valuable and very helpful for revising and improving our paper and have important guiding significance to our research. We will upload the response to the submission system. Please check.

---

## [Decision Letter · Decision Letter 2]

4 Apr 2023

Short-Term Power Load Forecasting Based on the CEEMDAN-TCN-ESN Model

PONE-D-23-03222R2

Dear Dr. Jiang,

We’re pleased to inform you that your manuscript has been judged scientifically suitable for publication and will be formally accepted for publication once it meets all outstanding technical requirements.

Kind regards,

Lin Wang, Ph.D.

Academic Editor

PLOS ONE

Additional Editor Comments (optional):

Reviewers' comments:

Reviewer's Responses to Questions

**Comments to the Author**

1. If the authors have adequately addressed your comments raised in a previous round of review and you feel that this manuscript is now acceptable for publication, you may indicate that here to bypass the “Comments to the Author” section, enter your conflict of interest statement in the “Confidential to Editor” section, and submit your "Accept" recommendation.

Reviewer #2: All comments have been addressed

2. Is the manuscript technically sound, and do the data support the conclusions?

Reviewer #2: Yes

3. Has the statistical analysis been performed appropriately and rigorously? 

Reviewer #2: Yes

4. Have the authors made all data underlying the findings in their manuscript fully available?

Reviewer #2: Yes

5. Is the manuscript presented in an intelligible fashion and written in standard English?

Reviewer #2: Yes

6. Review Comments to the Author

Reviewer #2: All of my comments are handled satisfactorily. I have no further questions. Now, it can be accepted.

7. PLOS authors have the option to publish the peer review history of their article (what does this mean?). If published, this will include your full peer review and any attached files.

Reviewer #2: No

---

## [Editor Report · Acceptance letter]

9 May 2023

PONE-D-23-03222R2 

Short-Term Power Load Forecasting Based on the CEEMDAN-TCN-ESN Model 

Dear Dr. Jiang:

I'm pleased to inform you that your manuscript has been deemed suitable for publication in PLOS ONE. Congratulations! Your manuscript is now with our production department. 

Kind regards, 

on behalf of

Dr. Lin Wang 

Academic Editor

PLOS ONE